# Unveiling the Dichotomy of Urinary Proteins: Diagnostic Insights into Breast and Prostate Cancer and Their Roles

**DOI:** 10.3390/proteomes12010001

**Published:** 2023-12-26

**Authors:** Yan Feng, Qingji Huo, Bai-Yan Li, Hiroki Yokota

**Affiliations:** 1Zhejiang Cancer Hospital, Hangzhou Institute of Medicine (HIM), Chinese Academy of Sciences, Hangzhou 310022, China; fengyan@zjcc.org.cn; 2Department of Pharmacology, College of Pharmacy, Harbin Medical University, Harbin 150081, China; qinghuo@iu.edu; 3Department of Biomedical Engineering, Indiana University Purdue University Indianapolis, Indianapolis, IN 46202, USA; 4Indiana Center for Musculoskeletal Health, Indiana University School of Medicine, Indianapolis, IN 46202, USA; 5Indiana University Simon Comprehensive Cancer Center, Indianapolis, IN 46202, USA

**Keywords:** urinary proteins, breast cancer, prostate cancer, physical activities, surgery, induced tumor-suppressing cells (iTSCs)

## Abstract

This review covers the diagnostic potential of urinary biomarkers, shedding light on their linkage to cancer progression. Urinary biomarkers offer non-invasive avenues for detecting cancers, potentially bypassing the invasiveness of biopsies. The investigation focuses primarily on breast and prostate cancers due to their prevalence among women and men, respectively. The intricate interplay of urinary proteins is explored, revealing a landscape where proteins exhibit context-dependent behaviors. The review highlights the potential impact of physical activity on urinary proteins, suggesting its influence on tumorigenic behaviors. Exercise-conditioned urine may emerge as a potential diagnostic biomarker source. Furthermore, treatment effects, notably after lumpectomy and prostatectomy, induce shifts in the urinary proteome, indicating therapeutic impacts rather than activating oncogenic signaling. The review suggests further investigations into the double-sided, context-dependent nature of urinary proteins, the potential role of post-translational modifications (PTM), and the integration of non-protein markers like mRNA and metabolites. It also discusses a linkage of urinary proteomes with secretomes from induced tumor-suppressing cells (iTSCs). Despite challenges like cancer heterogeneity and sample variability due to age, diet, and comorbidities, harnessing urinary proteins and proteoforms may hold promise for advancing our understanding of cancer progressions, as well as the diagnostic and therapeutic role of urinary proteins.

## 1. Introduction

Tumor biomarkers are detectable within bodily fluids like peripheral blood and urine [1,2,3,4,5,6]. Due to their direct involvement in urine generation or their close anatomical proximity to the urinary tract, cancers affecting organs like the bladder, kidney, prostate, and ovary stand to benefit from the utility of urinary biomarkers in diagnostic processes [2,7,8,9,10,11,12,13]. Urine biomarkers also hold diagnostic potential for breast cancer [14,15,16]. While urine contains cancerous cells or constituents derived from compromised cells, encompassing cellular debris and fragments of DNA, as well as diverse metabolites [17,18,19], this comprehensive review predominantly centers on urinary proteins in breast and prostate cancers, both being prominently diagnosed among women and men, respectively [20,21,22].

A specific question revolves around the contrasting roles that urinary proteins can assume in both promoting and inhibiting tumor growth. While serum protein markers employed for cancer detection are predominantly perceived as oncoproteins due to the synthesis and secretion of growth-promoting proteins by cancer cells, recent discoveries have unveiled the other side of perspectives [11,23]. We may consider interferon beta 1 (IFNβ1), which breast cancer cells secrete to reprogram stromal fibroblasts and foster tumor expansion [24,25]. Recent investigations, however, have highlighted the dual-faced nature of numerous oncoproteins contingent on the context [26,27]. This context-dependent behavior manifests in their potential to stimulate intracellular cancer cell proliferation, while simultaneously exhibiting tumor-suppressive qualities in their extracellular forms [28]. In the realm of breast cancer, enolase 1 (ENO1) stands as an illustrative example. As a glycolytic enzyme, ENO1 plays a significant role in cancer progression [29,30]. However, we have previously reported that its extracellular manifestation takes on a contrary role, functioning as a tumor suppressor [31]. This intricate interplay underscores the contextually driven, dual-faceted nature of proteins like ENO1, thereby paving the way for the identification of urinary proteins that can similarly act as both stimulators and suppressors of tumorigenesis.

Prompted by the potential dual actions of these proteins, a thought-provoking query arises: Could oncoproteins, commonly perceived and detected within urine, potentially serve as either agents of oncogenesis or as antidotes to oncogenesis? While the therapeutic merits of urine remain empirically unverified, the prospect of isolating anti-tumor proteins or peptides from urine stands as a notion not easily dismissed. Noteworthy is the identification, by us and other researchers, of tumor-suppressing proteins like serine protease 8 (PRSS8) and nidogen 1 (NID1) within the urine of prostate cancer patients [32,33,34,35]. The levels of these urinary proteins could undergo modifications contingent on external stimuli or therapeutic interventions. In the context of this assessment, our focus rests upon two illustrative examples: the impact of physical activities and surgical interventions, particularly lumpectomy for patients with breast cancer and prostatectomy for patients with prostate cancer. Both instances possess the potential to substantially reshape the profile of urinary proteins.

## 2. Urinary Biomarkers for Breast Cancer and Prostate Cancer

It has been debatable whether screening breast cancer and prostate cancer does more harm than good [36,37], whereas precise screening and diagnostics are recommendable in determining the suitable course of cancer treatment [38]. Early detection not only facilitates timely decision making but also augments both the quality of life and the overall survival rate [39]. Leveraging urinary constituents as diagnostic indicators proves especially advantageous for cancers intricately tied to the urinary tract, encompassing urothelial carcinoma, bladder cancer, and non-muscle invasive bladder cancer [5,39,40,41]. Beyond this domain of urinary tract-related cancers, urinary markers have demonstrated their utility in detecting breast cancer and prostate cancer (Table 1).

Breast cancer: Breast biopsies, a commonly employed method, entail invasiveness that can potentially expose patients to potential risks such as bruising, swelling, infection, and bleeding at the biopsy site [42]. In contrast, urine-based diagnostics present a non-invasive avenue [43]. A cluster of biomarkers is intertwined with the modulation of the extracellular matrix, exemplified by the matrix metalloproteinase 9 (MMP9) and neutrophil gelatinase-associated lipocalin (NGAL) complex [44].

According to Table 1, the elevated NGAL level correlates with the advancement of breast cancer, with NGAL-overexpressing tumors coinciding with heightened MMP9 levels. Comparative studies indicate that an escalated urinary MMP9 level corresponds to a fivefold risk of atypical hyperplasia and a more than thirteenfold risk of lobular carcinoma in situ (LCIS) compared to normal controls. Additionally, an augmented urinary concentration of disintegrin and metalloproteinase domain-containing protein 12 (ADAM12) is evident in women with atypical hyperplasia and LCIS [45].

Findings also indicate the significant elevation of a matrix metalloproteinase 1 (MMP1) and CD63 complex in the urine of breast cancer patients [46]. CD63, a cell surface molecule, binds with tissue inhibitor of metalloproteinases 1 (TIMP1). Furthermore, extracellular matrix protein 1 (ECM1), microtubule-associated serine/threonine kinase family member 4 (MAST4), and Filaggrin (FLG) exhibit heightened levels in the urine of breast cancer patients [47]. Apart from extracellular matrix proteinases, endothelial-derived gene 1 (EG1), expressed in both endothelial and epithelial cells, displays elevated levels not only in breast cancer but also in colon, prostate, and lung cancers [48]. Similarly, trefoil factor 1 (TFF1), a small secretory protein, demonstrates elevated levels across various cancer types, including breast cancer [30]. Moreover, ECM1, MAST4, FLG, and MAST4 are implicated as potential biomarkers for the preliminary indication of breast cancer presence [49].

Prostate cancer: At present, the serum prostate-specific antigen (PSA) stands as the foremost pivotal biomarker for discerning, tracking, and overseeing the treatment of prostate cancer [4,50,51,52,53]. Despite its instrumental role in substantially reducing prostate-cancer-related mortality, however, its utilization has also brought about the unintended consequences of excessive diagnosis and overtreatment of low-risk prostate cancer cases [54,55,56]. Consequently, an imperative exists for the development of more dependable, non-invasive approaches to prostate cancer diagnosis. Beyond PSA, another notable biomarker is prostate cancer antigen 3 (PCA3), marked by robust expression in individuals afflicted with prostate cancer [57]. This led to the FDA’s 2012 approval of PCA3’s use as a urine-based diagnostic tool for prostate cancer. Notably, the PCA3 level has been indicated to be independent of prostate size and serum PSA level [58,59,60]. Furthermore, within the urine of prostate cancer patients, the presence of Golgi membrane protein 1 (GOLM1) immunoreactivity has been identified, suggesting its potential as a biomarker for clinically localized prostate cancer [61].

In addition, Engrailed 2 protein (EN2), a transcription factor bearing a homeodomain, is secreted into the urine by prostate cancer cells, distinct from normal prostate tissue and benign prostatic hypertrophic cells that do not exhibit EN2 secretion [62]. The levels of urinary EN2 before radical prostatectomy have been linked to the stage of the tumor [63]. Another notable biomarker, TMPRSS2-ERG (V-ets erythroblastosis virus E26 oncogene homolog), fused with SAM-pointed domain-containing Ets-like factor (SPDEF), is recognized as a prostate-cancer-specific marker in urine [64,65]. Elevated SPDEF levels correlate with heightened aggressiveness and metastatic potential [48]. Furthermore, the urinary levels of β-2-microglobulin (β2M), pepsinogen A3 (PGA3), and mucin 3 (MUC3) were found to be elevated in prostate cancer patients [66]. Furthermore, urinary CD105 exhibited increased levels in men with biopsy-positive prostate cancer in comparison to those with biopsy-negative results [67]. The interleukin 18 binding protein (IL18BP), a potent inhibitor of IL-18, was also noted to be elevated in the urine of individuals with prostate cancer [47].

It is worth noting that urinary proteins are derived from blood filtration processes occurring in the kidneys. In cases where the filtration membrane of the kidneys is compromised by prostate cancer, it can result in the excretion of abnormal substances into the urine through this damaged filtration system. This condition, known as proteinuria, may arise due to various factors including kidney diseases and immune disorders [68,69].

**Table 1 proteomes-12-00001-t001:** Urinary biomarkers for breast cancer and prostate cancer.

**Breast Cancer**				
**Gene**	**Name**	**Function**	**FDA** **Approval ***	**Ref.**
MMP9 NGAL	matrix metalloproteinase 9	facilitating angiogenesis and tumor growth	Yes	[44]
	neutrophil gelatinase-associated lipocalin		Yes	
MMP1 CD63	matrix metalloproteinase 1	involved in the cancer development of breast cancer	Yes	[46]
	CD63		Yes	
MMP9 ADAM12	matrix metalloproteinase 9	urinary MMP9 and ADAM12 levels significantly increase with disease progression in breast cancer patients and correlate with the disease stage	Yes	[45]
	a disintegrin and metalloprotease 12		No	
EG1	endothelial-derived gene 1	EG1 stimulates cellular proliferation	No	[70]
TFF1	trefoil factor 1	the tumor grade was correlated with TFF1	No	[48]
ECM1 MAST4 Filaggrin	extracellular matrix protein 1	expression is highly significantly correlated with survival in breast cancer patients	No	[49]
	microtubule-associated serine/threonine kinase family member 4		No	
	filaggrin		No	
**Prostate Cancer**				
**Gene**	**Name**	**Function**	**FDA** **Approval ***	**Ref.**
PCA3	prostate cancer-associated 3	predict the tumor volume, extracapsular extension, and positive surgical margins in prostatectomy specimens	Yes	[58,71]
EN2	engrailed 2	pre-surgical urinary EN2 levels were associated with increasing tumor stage and closely reflected the volume of cancer in prostate cancer specimens	No	[62,63]
TMPRSS2-ERG SPDEF	transmembrane proteinase serine 2:v-ets erythroblastosis virus E26 oncogene homolog	predicting initial biopsy results in prostate cancer	No	[64,65]
	SAM-pointed domain-containing Ets-like factor		No	
β2M PGA3 MUC3	β-2-microglobulin	distinguish between benign prostate hyperplasia (BPH) and localized prostate cancer	No	[66]
	pepsinogen A3		No	
	mucin 3		No	
CD105	endoglin	urinary endoglin levels in men with prostate cancer correlated with radical prostatectomy tumor volume	No	[67]
IL18BP	interleukin-18 binding protein	IL18BP merits further study as a marker of aggressive prostate cancer and as a therapeutic target	No	[47]
GOLM1	Golgi membrane protein 1	GOLM1 is a resident cis-Golgi membrane protein of unknown function	No	[61]

* Searched for the list of qualified biomarkers [72].

## 3. Effects of Physical Activities

Physical activities have been known to be beneficial by strengthening bone and muscle and reducing inflammatory reactions, which are evidenced by serum proteome analyses [73,74,75,76,77]. Urine represents a filtrate of blood, thus rendering its protein constituents qualitatively akin to those present in the bloodstream. However, in comparison to serum proteins, urinary proteins tend to be more dilute and generally exhibit less complexity. Robust epidemiological evidence substantiates the protective impact of physical activity on breast cancer risk, recurrence, and mortality [78,79,80,81,82]. Research studies have elucidated that moderate exercise can distinctly enhance the prognosis of cancer patients by curbing tumor growth and forestalling metastasis [83,84,85,86,87]. Hydroxyproline serves as a prevalent urinary marker, indicative of the extent of connective tissue degradation encompassing bone, muscle, and other collagen and/or elastin-rich tissues [88,89,90]. Furthermore, research has indicated that physical exercises, particularly aerobic modalities, bring about a reduction in urinary liver-type fatty acid binding protein (L-FABP) levels, as well as a decrease in urinary albumin excretion [91,92,93].

It is reported that mice bearing mammary tumors that had access to running wheels displayed diminished growth in both MCF-7 and MDA-MB-231 tumors [94,95,96]. In a prior study of ours, alterations in urinary proteins were observed by collecting samples from mice subjected to 5 min tibia loading, as well as from human individuals before and after a 30 min session of step aerobics. In comparison to urine samples collected before these loading activities or step aerobics, post-activity urine exhibited a reduction in cellular viability, proliferation, migration, and invasion of tumor cells in cell culture investigations [97]. After the activity, post-activity urine exhibited a significant increase in dopamine and melatonin levels while concurrently decreasing cholesterol, a compound associated with tumor promotion (Figure 1).

Prior research has established that dopamine and melatonin can downregulate Lrp5, a co-receptor involved in Wnt signaling pathways [91,92,93]. On the contrary, cholesterol is known to upregulate Lrp5, aligning with the observed effects of urine on Lrp5 expression. Furthermore, individuals conditioned by aerobic exercise displayed a substantial reduction in CD105 levels in their urine. CD105 is positioned downstream of Lrp5 and CSF1, with the latter being a hematopoietic growth factor linked to bone homeostasis and the progression of various cancers [93]. Additionally, CD105 is a component of the TGFβ receptor complex, and its role extends to tumor-associated angiogenesis [94]. These findings collectively contribute to the suppression of genes known to promote tumorigenesis, such as Snail, MMP9, Runx2, and PPARγ, within tumor cells. Additionally, administering diluted post-activity urine samples via intraperitoneal injection led to decreased tumor weight in the mammary fat pad within a mouse model of breast cancer [97]. These outcomes collectively underscore the potential of loading-conditioned urine not only as a prospective tumor suppressor but also as a wellspring of diagnostic biomarkers [98,99,100].

Aerobic exercise can also reduce the adverse effects of prolonged sitting, hypertension, and interstitial damage in patients with chronic kidney disease, by decreasing the level of urinary liver-type fatty acid binding protein (L-FABP) [91]. The level of urinary alkaline phosphatases is also reported to change in young men before and after 3 km running [101]. Similarly, swimming exercises can alter urine proteomics [102]. A study has been conducted to investigate the effects of long-distance cycling on specific urinary biomolecules [103]. The participants exhibited significant increases in the levels of serum lactate, uric acid, and bilirubin, even though they are not proteins. Notably, uric acid plays a crucial role in vascular regulation by increasing oxidative stress and promoting nitric oxide clearance, thereby inducing vasodilation [104]. An increase in bilirubin levels may contribute to a reduction in cardiovascular risk [105]. In summary, the growing body of evidence supports the beneficial impact of physical activity on both urinary proteins and non-protein biomolecules.

## 4. Treatment Effects

As surgical treatment affects urinary proteomes, two studies are reported for lumpectomy for breast cancer and prostatectomy for prostate cancer. When compared to healthy subjects, a noteworthy increase in the concentration of urinary protein ADAM 12 was observed in patients with breast cancer who underwent lumpectomy. While the reported study cannot definitively establish a direct connection between the concentration of urinary protein ADAM 12 and the status and stage of breast cancer, it does suggest that surgical tumor resection has an impact on urinary protein [106]. Additionally, in the case of urine samples collected from patients with breast cancer following surgery or other treatments, a correlation has been established between urinary estrogen metabolism levels and breast cancer risk [107].

Regarding prostatectomy, distinct discrepancies in urinary proteomes were discerned between two groups of prostate cancer patients: those with positive surgical margins and those with negative margins. Notably, the positive margin group exhibited elevated levels of three proteins—cyclin-dependent kinase 6, galectin-3-binding protein, and L-lactate dehydrogenase C chain [108]. In a separate study, urine samples were procured from patients with prostate cancer and breast cancer, all of whom underwent external beam radiation therapy, without concurrent chemotherapy. The findings illuminated elevated levels of VEGF and MMP in patients with cancer compared to those in normal controls [109,110]. Furthermore, individuals with metastatic cancer displayed even higher VEGF and MMP levels when contrasted with patients diagnosed with non-metastatic cancer [109].

One of the primary objectives is to harness urinary proteins as a dual-purpose tool—both for diagnosis and as an indicator of the effectiveness of treatments like chemotherapy, radiotherapy, and surgery. Our earlier research unveiled distinctive shifts in the urinary proteomes of prostate cancer patients, showcasing variable levels of tumor-modulating proteins in urine samples taken before and after prostatectomy (Figure 2). Initially, we found that applying diluted urine obtained from patients after prostatectomy, a procedure entailing the surgical removal of the prostate led to a substantial reduction in the tumorigenic behaviors exhibited by prostate tumor cells. In post-prostatectomy urine, the levels of angiogenin [32,111]—a promoter of blood vessel formation—were significantly diminished [32]. Additionally, the post-prostatectomy urine demonstrated heightened concentrations of three cell-membrane proteins: PRSS8 [112], nectin 2 (PVRL2) [113], and NID1 [114]. Notably, these proteins exerted tumor-suppressive actions within the extracellular domain by inhibiting the expression of oncogenic genes such as Snail and TGFβ.

Of significant importance, NID1 emerges as a multifaceted protein with a dual role, whereby its functional impact varies based on its cellular location. In its extracellular milieu, NID1 exhibits tumor-suppressive qualities [115], while its intracellular presence potentiates tumor-promoting effects [116,117]. This dichotomy is notably demonstrated in the context of cell migration, where the introduction of recombinant NID1 or conditioned medium from cells overexpressing NID1 led to diminished migratory behavior in triple-negative breast cancer. These observations suggest that NID1’s role as a tumor suppressor could potentially be harnessed as a valuable therapeutic avenue for the treatment of triple-negative breast cancer.

Utilizing quantitative-mass-spectrometry-based proteomic analysis, it was unveiled that NID1 proteins are secreted by endothelial cells, exerting an inhibitory effect on the migration of cancer cells induced by endothelial cells [115]. However, it is noteworthy that NID1’s association with ovarian cancer reveals a contrasting facet. In this context, NID1 contributes to a poor prognosis by promoting invasion, migration, and chemoresistance in ovarian cancer through the activation of ERK/MAPK signaling [114]. On a related note, the role of the serine protease PRSS8 emerges as a potential suppressor in colorectal carcinogenesis and metastasis [118,119,120]. Its ectopic expression has demonstrated the capability to inhibit tumor growth both in vitro and in vivo, in addition to curbing the migration and invasion of non-small-cell lung cancer cells [121].

An advantage of using urine as a medium over blood is the relative stability of urinary proteins, as they do not undergo significant proteolysis within several hours of collection. Therefore, urinary proteomics offers an attractive avenue for the discovery of cancer biomarkers [122]. In the urinary proteomes of human prostate cancer specimens obtained after prostatectomy, variations have been observed between groups of patients with positive and negative surgical margins [108]. These differences can be linked to the underlying molecular mechanisms of prostate cancer development [108].

## 5. Double-Sided Role of Urinary Proteins

Not only urinary proteins but also some tumor suppressor proteins are double-sided and very dependent on the environment. Some oncogenic proteins in the cytoplasm and cell membrane are thought to promote tumor cell proliferation and migration but may conversely act as tumor suppressor proteins in the extracellular domain. For instance, extracellular Eno1 recombinant proteins are reported to suppress the metabolic activities of breast cancer cells and act as cytotoxic agents by downregulating Snail, TGFβ, and MMP9 [123]. By contrast, the overexpression of Eno1 in breast cancer cells upregulated the above tumorigenic genes and elevated their proliferation and transwell invasion [124]. The protein NID1 in urine and its recombinant protein reduced the EdU-based proliferation and scratch-based motility of TRAMP prostate cancer cells. In contrast, overexpression of NID1 stimulated the EdU-based proliferation and scratch-based motility of TRAMP cells. This shows that extracellular NID1 acts as a tumor suppressor gene, while intracellular NID1 acts as a tumor-promoting gene [32]. In addition, urinary protein CD14 inhibits gastric cancer cell invasion and epithelial-mesenchymal transition after knockout in gastric cancer cells MGC-803, and our previous study showed that recombinant protein CD14 can significantly inhibit the proliferation and invasion of TRAMP cells outside cells [32,125].

An intriguing question arises when considering the dual role of urinary proteins and their potential connection to proteoforms, which denote the diverse molecular forms of proteins. It remains uncertain whether a protein’s functionality is influenced solely by distinct cellular locations, such as cytoplasmic and extracellular domains, or if variations in functions, like those in cell signaling, cell adhesion, and interactions, stem from modifications, cleavage, and other alterations. Addressing this question requires further studies to unravel the intricate relationships between urinary proteins, proteoforms, and their multifaceted roles. It is important to note that the double-sided role of urinary proteins may differ depending on not only their locations in the cytoplasm, extracellular space, or urine but also the distinctive microenvironments associated with cancer and interacting cells. The role in the urine can also be affected by age, diet, and hydration status. Given these complexities, further analyses are recommended to characterize their value as diagnostic and prognostic tools.

## 6. Future Perspective

To comprehensively unravel the intricate role of urinary proteins in individuals afflicted with breast and prostate cancer, it is advisable to delve into the following four avenues of investigation (Figure 3).

Comparison with Serum Components: Exploring the parallelisms and divergences between urinary and serum proteins holds immense potential [126,127]. Establishing both the similarities and disparities could shed light on the unique attributes of urinary proteins and enable a more comprehensive understanding of their diagnostic and prognostic significance. An essential aspect of this endeavor involves standardizing the collection methodologies and ensuring consistency and reliability in the data obtained from serum and urine samples.

Dual-Faced Functionality: Recognizing the dual roles of urinary proteins—whether they act as oncogenic agents or antioncogenic factors like tumor-suppressing proteins from iTSCs—requires meticulous examination [128]. This investigation would entail deciphering the intricate interplay of these proteins in different contexts and disease stages. By delineating the mechanisms through which they can either stimulate or inhibit tumorigenesis, a clearer picture of their contributions to cancer development and progression can be formed.

Proteoform Changes: Understanding the diverse proteoforms of urinary proteins holds critical importance in the realms of biomarker discovery, disease diagnostics, and comprehending the dynamic nature of the urinary proteome. Proteoform alterations, encompassing post-translational modifications (PTMs), the cleavage of larger proteins into smaller subunits, and amino acid sequence variants, play a pivotal role in distinguishing both normal and pathological conditions. PTMs can occur shortly after translation or at any stage in a protein’s lifecycle, contributing significantly to our functional understanding of biology. Phosphorylation, one of the most extensively studied modifications, can dynamically and rapidly regulate various signaling pathways in both health and disease conditions [129].

Studies report altered patterns of protein tyrosine phosphorylation in the urine of patients with bladder cancer [130]. Additionally, disease-associated modifications in the glycosylation of urinary proteins have been well documented [131], with anomalies in glycosylation linked to the development and progression of malignant tumors. Notably, research has indicated the presence of sialylated N-glycans in PSA found in the serum of patients with prostate cancer [132]. Despite urine being a widely accessible biological sample, easily obtained for analysis, there remains a scarcity of PTM information, such as phosphorylation, glycosylation, acetylation, and SUMOylation, for urine protein analysis [133]. The development of phosphorylation information related to urine proteins in patients with breast and prostate cancers could pave the way for personalized medical approaches.

Beyond the protein modifications such as phosphorylation and glycosylation, the dynamic world of urinary proteins extends to a fascinating realm of cleavage, giving rise to fragmented proteoforms with potential implications as antigens or pro-/anticancer peptides. Notably, there is an ongoing exploration of peptides derived from tumor antigens, presenting a promising avenue for the advancement of peptide-based cancer vaccines [134,135]. Moreover, a noteworthy stride in anticancer research involves the synthesis of peptides sourced from the secretome of both cancer cells and iTSCs [136,137]. It is imperative to delve into the intricate landscape of these urinary protein fragments, particularly the smaller subunits, intending to unlock their full potential for diagnostic and therapeutic applications. The potential utility of these subunits in precision medicine, as well as their role in unraveling novel pathways for targeted therapies, underscores the significance of advancing our understanding and application of urinary proteins. Thus, a concerted effort to unravel the complexities and nuances of these urinary protein fragments is recommended, offering a promising frontier for future breakthroughs in diagnostics and therapeutics.

Beyond Protein Markers: While proteins are vital players, delving into the realm of non-protein markers can broaden our understanding. The inclusion of markers like mRNA [138,139], metabolites [140,141], and volatile organic compounds (VOC) [39,142,143] offers a multi-dimensional perspective on cancer-related alterations. Exploring these components in tandem with urinary proteins could provide a more holistic view of the molecular signatures associated with breast and prostate cancer. This comprehensive approach might uncover novel diagnostic, prognostic, and therapeutic insights, potentially enhancing patient care.

In sum, through an integrated investigation spanning comparisons with serum components, an exploration of dual roles, and the incorporation of non-protein markers, a deeper comprehension of the intricate role of urinary proteins in breast and prostate cancer patients can be attained. It is highly desirable to identify biomarkers that could be monitored in urine for the follow-up of cancer progression, their role in survival, diagnosis, response, and changes due to the effect of interventions.

## 7. Limitations of This Review

Amidst the intricate array of constituents found within the urine, the urinary proteome remarkably lends itself to clinical investigation due to several key advantages. These advantages stem from the abundant availability of urine samples, the noninvasive methodology of collection, and the potential for repeated sampling. The study of urinary proteomes offers significant advantages for clinical research, yet the complexities associated with sample variability, proteome intricacies, diagnostic specificity, and sample standardization warrant careful consideration.

Several reports highlight potential challenges and limitations in analyzing urinary proteomes. One significant consideration is the influence of diet, specifically diet-induced hypercholesterolemia, which has been observed to affect the expression of oxidized low-density lipoprotein [144]. Notably, oxidized low-density lipoprotein has been identified as a potential biomarker for bladder cancer. Furthermore, age dependence represents another constraint. For example, while S100 calcium-binding protein A16 (S100A16) holds promise as a prognostic biomarker for bladder cancer, its expression is associated with age, recurrence rates, and cancer-specific mortality [145]. It is also important that the biomarkers for cancer should be distinguished from those for common urinary problems such as bladder infections and kidney diseases.

Despite these limitations, harnessing the potential of urinary proteins provides a valuable avenue for advancing our understanding of diseases like breast and prostate cancer. The advancing field of comprehensive genetic analysis, made possible by the strides in next-generation sequencing, could significantly impact the future utilization of urinary markers [146].

## 8. Conclusions

Urine composition, influenced by physiological factors and external stimuli like physical activity, holds crucial diagnostic information, especially for breast and prostate cancer. Examining urinary proteomes reveals a dual-role landscape, with proteins exhibiting both tumor-promoting and tumor-suppressive roles. In breast cancer, non-invasive detection through urinary biomarkers, such as MMP9, NGAL, CD63, ECM1, MAST4, and Filaggrin, offers insights into disease progression. For prostate cancer, PSA, PCA3, GOLM1, and EN2 serve as valuable markers, highlighting diagnostic and prognostic potential. The impact of physical activity on urinary proteins, including loading-conditioned urine, shows promise as a tumor suppressor and diagnostic source. Post-lumpectomy and prostatectomy-induced shifts in the urinary proteome also indicate the potential for these proteins to reflect treatment effects. Further investigations comparing urinary and serum components, understanding the dual roles of urinary proteins, and incorporating non-protein markers can deepen insights into their role in cancer. Despite challenges, harnessing urinary proteins holds the potential to enhance disease understanding and patient care. Acknowledging the inherent heterogeneity and variations arising from factors like age and diet, the utilization of urinary proteins holds the potential to enhance our understanding of cancers and improve patient care.

## Figures and Tables

**Figure 1 proteomes-12-00001-f001:**
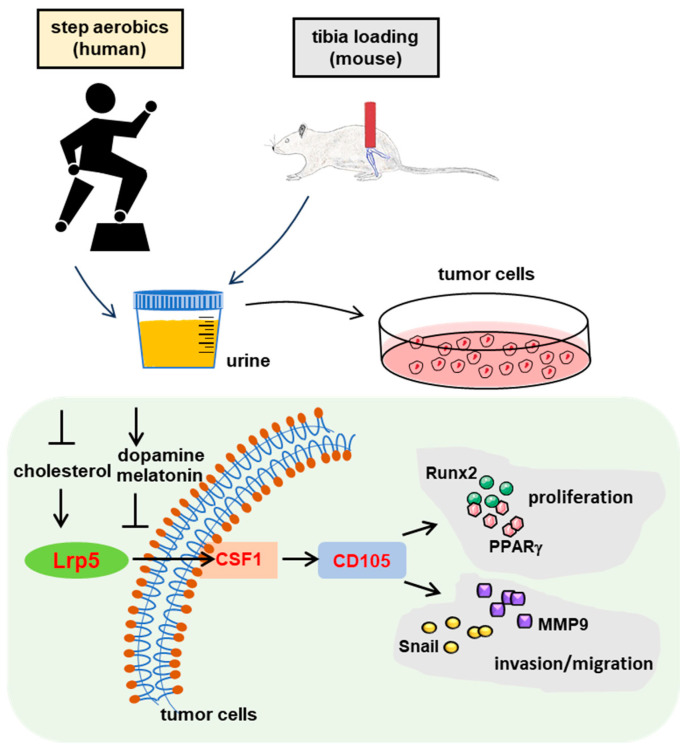
Proposed regulatory mechanism of the action of loading/aerobics-conditioned urine. Tibia loading in mice elevates urinary levels of tumor-suppressing dopamine and melatonin, simultaneously reducing tumor-promoting cholesterol. Similarly, in humans, activities like step aerobics yield these effects, inhibiting tumor-promoting genes such as Snail, MMP9, and Runx2 in tumor cells through the downregulation of Lrp5. Of note, Lrp5: low-density lipoprotein receptor-related protein 5, CSF1: macrophage colony-stimulating factor 1, CD105: endoglin, MMP9: matrix metalloproteinase 9, Runx2: runt-related transcription factor 2, and PPARγ: peroxisome proliferator-activated receptor gamma.

**Figure 2 proteomes-12-00001-f002:**
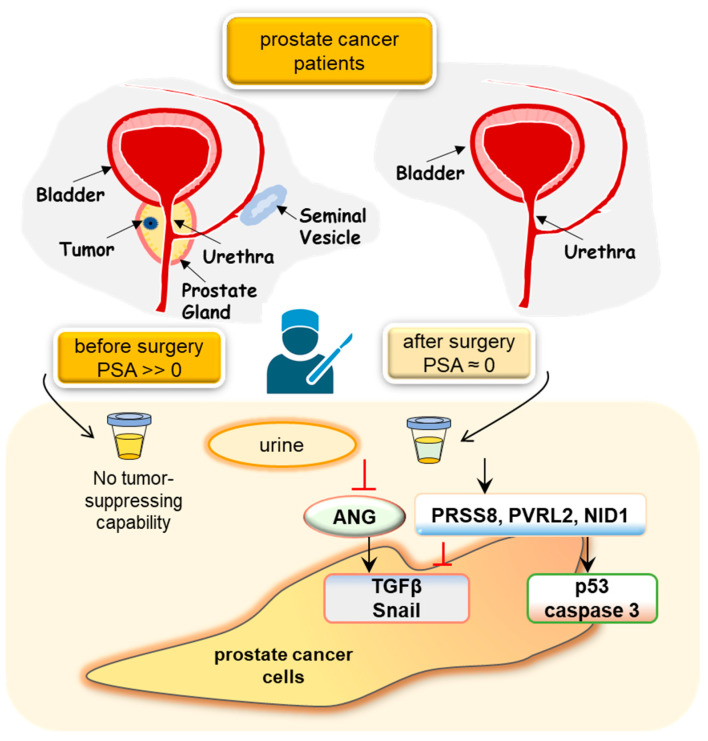
Putative tumor-suppressing mechanism by urine from the post-prostatectomy patients. Post-prostatectomy patient urine samples exhibit higher concentrations of PRSS8, PVRL2, and NID1. These elements, recognized as tumor suppressors, effectively decrease the presence of tumor-promoting proteins like TGFβ and Snail. Simultaneously, they enhance the expression of p53 and c-caspase 3, thereby triggering the apoptosis of prostate cancer cells. Of note, ANG: angiogenin, PRSS8: prostasin, PVRL2: nectin 2, and NID1: nidogen 1.

**Figure 3 proteomes-12-00001-f003:**
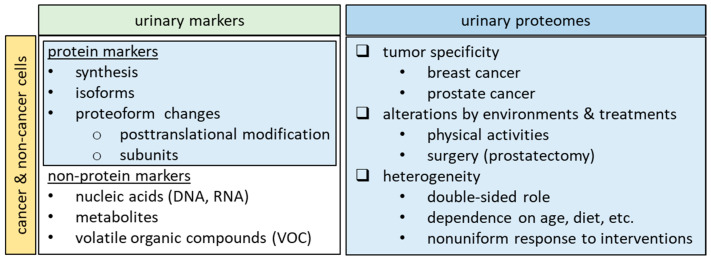
Unraveling the intricate role of urinary protein from urinary markers and urinary proteomes in tumor and non-tumor cells.

## Data Availability

Not applicable.

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
