# Peer review of "Unveiling the Dichotomy of Urinary Proteins: Diagnostic Insights into Breast and Prostate Cancer and Their Roles"

_proteomes, 2023, doi:10.3390/proteomes12010001_

Round 1

Reviewer 1 Report

Comments and Suggestions for Authors

The article "Unveiling the Dichotomy of Urinary Proteins: Diagnostic Insights into Breast and Prostate Cancer and Their Roles" is fascinating. This review article evaluates urine-derived proteins that have a dual role in cancers such as prostate and breast cancer and highlights the potential impact of physical activity on urinary proteins.

However, I believe several observations should be considered:

The article handles highly descriptive information; I suggest that it should be organized so that the following could be considered:

Biomarkers described in urine could potentially be associated with various cancers.

Probably evaluate whether there are markers that have been commonly described among the cancers that the review considers.

Studies could be organized according to proteomic analyses in patient samples or animal models, covering that dichotomy criterion. Then, functional analyses could suggest their role in both types of cancer and end with those urinary markers that have been observed to be modified in terms of abundance, subcellular localization, and PTM, among others, in the different types of cancer.

The search for biomarkers that could be monitored in urine for the follow-up of cancer progression and their role in survival, diagnosis, response, and changes due to the effect of interventions seems to me highly desirable.

An integrative image correlating such relevance would be of great help.

Finally, is exercise the only intervening factor that could modify these markers? Some other factors associated with cancer, such as age, comorbidities, diet, etc., could not affect these markers.

Comments on the Quality of English Language

some words seem to me to be stilted

Author Response

The article "Unveiling the Dichotomy of Urinary Proteins: Diagnostic Insights into Breast and Prostate Cancer and Their Roles" is fascinating. This review article evaluates urine-derived proteins that have a dual role in cancers such as prostate and breast cancer and highlights the potential impact of physical activity on urinary proteins. However, I believe several observations should be considered:

The article handles highly descriptive information; I suggest that it should be organized so that the following could be considered: Biomarkers described in urine could potentially be associated with various cancers. Probably evaluate whether there are markers that have been commonly described among the cancers that the review considers.

Thank you for the comment. In the revision, we added specific information such as the current status of FDA approval for the list of proteins in Table 1. As suggested, some biomarkers are shared with multiple cancers. We added more information on the biomarkers common to multiple cancers. Particularly, we added a description of bladder cancer that is closely linked to urine.

Studies could be organized according to proteomic analyses in patient samples or animal models, covering that dichotomy criterion. Then, functional analyses could suggest their role in both types of cancer and end with those urinary markers that have been observed to be modified in terms of abundance, subcellular localization, and PTM, among others, in the different types of cancer.

Thank you for the comment. Based on the suggestion, we slightly changed the order of the descriptions. In the effects of surgery, we started introducing two clinical observations, one for lumpectomy for patients with breast cancer and the other for prostatectomy for patients with prostate cancer, followed by the description of urinary markers.

The search for biomarkers that could be monitored in urine for the follow-up of cancer progression and their role in survival, diagnosis, response, and changes due to the effect of interventions seems to me highly desirable. An integrative image correlating such relevance would be of great help.

Thank you for the comment. In the revision, we added the suggested description, “it is highly desirable to identify biomarkers that could be monitored in urine for the follow-up of cancer progression, their role in survival, diagnosis, response, and changes due to the effect of interventions.” We also modified Figure 3 to better fit the added description.

Finally, is exercise the only intervening factor that could modify these markers? Some other factors associated with cancer, such as age, comorbidities, diet, etc., could not affect these markers.

Thank you for the comment. In the revision, we added a description of the other factors associated with age, diet, comorbidities, etc.

Comments on the Quality of English Language: some words seem to me to be stilted.

Thank you for the comment. We requested proofreading by a native writer.

Reviewer 2 Report

Comments and Suggestions for Authors

Overall, I find the review to be quite comprehensive, offering valuable insights into the potential future use of urinary biomarkers in cancer diagnosis and monitoring. Additionally, the review sheds light on the potential impact of physical activity on urinary proteins as well as their treatment effects.

Minor suggestions:

 1. Enhancing readability by using commonly understood words is recommended. While acknowledging the author's precise word choices, in a review article, readability is important for the audience. Using more commonly used terms might facilitate better understanding. For instance, replacing phrases like "captivating inquiry," "nuanced perspective", "instigator", and “akin to a double-sided moonlighter” with simpler language is recommended.

2. Could the author please consider annotating or adding a column in Table 1 that specifies whether the markers mentioned have received FDA approval? For instance, as highlighted by the author, PCA3 has already gained approval. This addition would enhance clarity and better demonstrate the potential of the urinary markers.

3. For the Figure legend, kindly relocate the annotation of proteins to the end of the legend instead of in the middle. For example, in Figure 2, move ANG: angiogenin, PPSS8: prostasin, PVRL2: nectin 2, and NID1: nidogen after the sentences “…apoptosis of prostate cancer cells” in the legend.

4. Line 254: “Therefore, urinary proteomics offers an attractive avenue…” However, later in this paragraph, the author begins discussing the utilization of urinary estrogen metabolism levels for breast cancer, which may require revision to better support the author's points regarding the advantages of urinary proteomics.

5. More discussion of limitations is recommended, or the authors could guide the readers toward other related reviews outlining the challenges of urinary proteomics analysis in clinical settings.

Author Response

Overall, I find the review to be quite comprehensive, offering valuable insights into the potential future use of urinary biomarkers in cancer diagnosis and monitoring. Additionally, the review sheds light on the potential impact of physical activity on urinary proteins as well as their treatment effects.

Minor suggestions:

  1. Enhancing readability by using commonly understood words is recommended. While acknowledging the author's precise word choices, in a review article, readability is important for the audience. Using more commonly used terms might facilitate better understanding. For instance, replacing phrases like "captivating inquiry," "nuanced perspective", "instigator", and “akin to a double-sided moonlighter” with simpler language is recommended.

Thank you for the comment. As suggested, we corrected the description.

  1. Could the author please consider annotating or adding a column in Table 1 that specifies whether the markers mentioned have received FDA approval? For instance, as highlighted by the author, PCA3 has already gained approval. This addition would enhance clarity and better demonstrate the potential of the urinary markers.

Thank you for the comment. As suggested, we added a column to Table 1 specifying whether the markers mentioned are FDA-approved.

  1. For the Figure legend, kindly relocate the annotation of proteins to the end of the legend instead of in the middle. For example, in Figure 2, move ANG: angiogenin, PPSS8: prostasin, PVRL2: nectin 2, and NID1: nidogen after the sentences “…apoptosis of prostate cancer cells” in the legend.

Thank you for the comment. As suggested, we corrected the description.

  1. Line 254: “Therefore, urinary proteomics offers an attractive avenue…” However, later in this paragraph, the author begins discussing the utilization of urinary estrogen metabolism levels for breast cancer, which may require revision to better support the author's points regarding the advantages of urinary proteomics.

Thank you for the comment. As suggested, we reorganized the description. In the revised manuscript, the description of lumpectomy for patients with breast cancer was placed at the beginning of subsection 4 (treatment effects), including the estrogen metabolism level. By rearranging the position of this paragraph as suggested, the subsection is now organized better. Thank you.

  1. More discussion of limitations is recommended, or the authors could guide the readers toward other related reviews outlining the challenges of urinary proteomics analysis in clinical settings.

Thank you for the comment. As suggested, we added a description of the limitations, including the lack of information on age, diet, and comorbidities. We also cited three review articles that are related to urinary proteomics analysis in bladder cancer.

  • Tomiyama E et al. (2023). Urinary markers for bladder cancer diagnosis: a review of current status and future challenges. Int J Urol. PMID:37968825.
  • Maas M et al. (2023). Urine biomarkers in bladder cancer – current status and future perspectives. Nature Rev Urol 20:597-614.
  • Ecke T (2015). Biomarker in Cisplatin-based chemotherapy for urinary bladder cancer. Adv Exp Med Biol 867:293-316.

Reviewer 3 Report

Comments and Suggestions for Authors

Thank you for allowing me to review this article.

The authors summarized the functions of urinary proteins in prostate cancer and breast cancer patients well. The specific protein names and their roles were listed, and their usefulness as biomarkers for diagnosis and treatment efficacy was described clearly and comprehensively. 

These contents seem to be meaningful to readers.

The following is the specific comment I would like the authors to explain.

-In prostate cancer, are cancer-related urinary proteins filtered out of the blood by the kidney or secreted from the prostate directly?

Author Response

Thank you for allowing me to review this article.

The authors summarized the functions of urinary proteins in prostate cancer and breast cancer patients well. The specific protein names and their roles were listed, and their usefulness as biomarkers for diagnosis and treatment efficacy was described clearly and comprehensively. These contents seem to be meaningful to readers. The following is the specific comment I would like the authors to explain.

In prostate cancer, are cancer-related urinary proteins filtered out of the blood by the kidney or secreted from the prostate directly?

Thank you for the comment. We added a description of the role of the kidney in cancer-related urinary proteins. In the description, we cited the following two references.

  • Shahinian VB, Bahl A, Niepel D, Lorusso V (2017). Considering renal risk while managing cancer. Cancer Manag Res. 9:167-178.
  • Swensen, Adam C et al. (2021). A Comprehensive Urine Proteome Database Generated From Patients With Various Renal Conditions and Prostate Cancer. Frontiers in Medicine 8:548212.

Reviewer 4 Report

Comments and Suggestions for Authors

The authors provided a summary of some possible urinary biomarkers for breast and prostate cancer, which are the two most prominently diagnosed cancers among women and men. While this topic is of great importance, this review does not provide any new additional information on the reliability of biomarkers identified as urinary biomarkers for the follow-up of cancer progression, aside from PCA3. Additionally, they introduce the idea that exercise and treatment can influence the expression level of these biomarkers in urine without much concrete evidence. While exercise or treatment has been shown to induce differential expression of some of these biomarkers, it is a bit of an outstretch at the present time to establish causality, certainly in urine, as they mentioned clearly in their conclusion. They further complicated their review by discussing proteoforms as well as non-protein markers such as mRNA and metabolites, both of which could also be valuable biomarkers in cancer detection and progression.

In conclusion, while the review addresses important topics, it is unclear what specific research question the authors are attempting to answer, despite the fact that a lot of factors can influence the expression of these biomarkers. As a result, I believe the authors' study question should be more specific. For example, they could limit their study to either breast or prostate cancer and focus on one of the biomarkers that could be a reliable urinary biomarker. Furthermore, they might emphasise the importance of possible parameters influencing its level of expression in urine, such as exercise, and, as a result, the need to consider the overall clinical setting. In conclusion, I believe that the review in its present form is of little interest to the community.

Author Response

The authors provided a summary of some possible urinary biomarkers for breast and prostate cancer, which are the two most prominently diagnosed cancers among women and men. While this topic is of great importance, this review does not provide any new additional information on the reliability of biomarkers identified as urinary biomarkers for the follow-up of cancer progression, aside from PCA3. Additionally, they introduce the idea that exercise and treatment can influence the expression level of these biomarkers in urine without much concrete evidence. While exercise or treatment has been shown to induce differential expression of some of these biomarkers, it is a bit of an outstretch at the present time to establish causality, certainly in urine, as they mentioned clearly in their conclusion. They further complicated their review by discussing proteoforms as well as non-protein markers such as mRNA and metabolites, both of which could also be valuable biomarkers in cancer detection and progression.

Thank you for the insightful comment. Regarding new additional information on the reliability, we added a status of FDA approval in Table 1. Regarding the effect of exercise and treatment, we revised it to avoid any overstatement acknowledging the limitations and emphasizing the cautious interpretation of a double-sided role of urinary proteins. Regarding proteoforms and nonprotein markers, they are not the focus of this review. However, we believe that their analysis is useful in evaluating the unique role of urinary proteins and protein variations.

In conclusion, while the review addresses important topics, it is unclear what specific research question the authors are attempting to answer, despite the fact that a lot of factors can influence the expression of these biomarkers. As a result, I believe the authors' study question should be more specific. For example, they could limit their study to either breast or prostate cancer and focus on one of the biomarkers that could be a reliable urinary biomarker. Furthermore, they might emphasise the importance of possible parameters influencing its level of expression in urine, such as exercise, and, as a result, the need to consider the overall clinical setting. In conclusion, I believe that the review in its present form is of little interest to the community.

Thank you for the comment. We focus on breast and prostate cancers, with a specific question as to whether urinary proteomes may present a double-sided role, which can be affected by exercise and treatment, as well as a list of existing and potential biomarkers. We appreciate your suggestion to emphasize uncontrolled parameters influencing biomarker expression in urine. We also understand a concern about the perceived limited interest in the community in the current form.

In the revision, we added the status of FDA approval in Table 1. We also rearranged two treatment examples, lumpectomy for breast cancer and prostatectomy for prostate cancer. We added some information on bladder cancer and urinary proteome, as the other reviewer requested to do so. To avoid any misinterpretation, we included a description of heterogeneity and variations of urine samples that are linked to age, diet, and comorbidities.

Round 2

Reviewer 1 Report

Comments and Suggestions for Authors

no additional comments

Author Response

Thank you!

Reviewer 4 Report

Comments and Suggestions for Authors

The authors have improved the manuscript by adding limitations to the use of urine biomarkers that need to be considered. 

One more suggestion for the authors: could you please include in your reference list the FDA webpages for the list of qualified biomarkers (breast and prostate) as well as your access date, both of which might be relevant for future publication on the subject?

Author Response

Thank you for the comment. As suggested, we included the FDA website on the reference list with the access date (new reference #72).